# The Association of Technology-Based Ageism with Using Digital Technology in Physical Therapy for Older Persons

**DOI:** 10.3390/healthcare11192672

**Published:** 2023-10-02

**Authors:** Cynthia Neiertz, Eveline J. M. Wouters, Ittay Mannheim

**Affiliations:** 1School of Allied Health Professions, Fontys University of Applied Science, 5631 BN Eindhoven, The Netherlands; cynthia.neiertz@yahoo.de (C.N.); e.wouters@fontys.nl (E.J.M.W.); 2Tranzo, School of Social and Behavioural Sciences, Tilburg University, 5037 AB Tilburg, The Netherlands; 3Department of Communication, Ben-Gurion University of the Negev, P.O. Box 653, Beer-Sheva 8410501, Israel

**Keywords:** digital technology, ageism, older adults, physiotherapists, healthcare

## Abstract

Integrating digital technologies in healthcare for older adults can enhance their independence and quality of life. Nevertheless, ageism among healthcare professionals impacts treatment decisions and may deprive older patients of technology-based treatment. This study explores whether technology-specific ageism influenced physiotherapists’ use of technology-based healthcare with older patients. Seventy-eight physiotherapists in Luxembourg filled out an online survey. Participants filled out the Attitudes Towards Older Adults Using Technology (ATOAUT-11) scale, Expectations Regarding Aging, attitudes towards technology use in the work environment, and whether they had not offered technology-based treatment in the past because of a patient’s age. Using logistic regression, negative ATOAUT was found to predict not offering technology-based treatment, such that participants with more negative attitudes (1 standard deviation) were two times more likely not to offer treatment. Positive attitudes towards using technology in the work environment were also found to be a significant predictor. All other characteristics (gender, age, experience and percentage of patients over 50) were not predictive of not offering treatment. This study demonstrates that technology-specific ageism may lead to discrimination and deprive older persons of optimal treatment. More research is needed to identify the magnitude of ageism in using technology-based treatment and develop interventions to overcome it.

## 1. Introduction

In the context of an ageing population, a common discourse is that implementing Digital Technology (DT) in the healthcare environment can help older persons maintain an independent life for a longer time in a safer way, as well as increase their quality of life and support healthcare [1,2,3]. However, recent studies point out that older adults are stereotypically seen as a group of persons who are incapable of using DT and not as individuals with diverse needs and capabilities [4,5,6]. This stereotypical view of older persons and manifestations of ageism is also present among healthcare professionals [7] and might negatively influence their willingness to use DT in treatment with older adults [8,9].

Ageism consists of three components, namely a cognitive component (stereotypes), an emotional component (prejudice), and a behavioural component (discrimination) [10]. Studies have shown that ageism is omnipresent in our society [11,12] and that negative thoughts and beliefs about older adults often happen unconsciously [13]. Ageist thoughts of healthcare professionals about older adults might also negatively influence the care they provide in terms of quality and quantity [14,15,16].

Besides the discriminatory aspect of ageism in healthcare, Meisner [17] suggests that the increasing prevalence of general ageism in the context of DT and the intergenerational tensions on social media exacerbated by the COVID-19 pandemic have increased negative stereotypes and discrimination towards older individuals, who may struggle to keep up with the fast pace of technological advancements. Moreover, a study by Choi et al. [18] found that perceived ageism can also widen the digital divide by further marginalising older individuals who may already struggle with accessing and using DT. The study highlights that this issue may be further compounded by gender, as older women may be more likely to experience ageism in their interactions with DT [18]. Thus, the presence of ageism in the context of DT may lead to older individuals being unable to fully participate in a digital society, including digital applications of healthcare.

The high prevalence of ageism in the healthcare system may influence which treatment a patient will receive. In their systematic review, Chang et al. [16] found that 84.6% of the studies they included showed that chronological age dictates which treatment or procedure will be chosen, meaning that a younger patient has a better outlook on receiving the gold standard treatment. For example, the systematic review looks at patients awaiting breast cancer surgeries and found in multiple studies that older patients were much more likely to be rejected for surgical treatment than younger patients with identical histories [16]. In the context of physiotherapy, Ambady et al. [19] discovered that non-verbal communication of physiotherapists during treatment, such as distancing themselves or looking away from the patient, was associated with physical decline of the older patient, both in the short term and the long term. They proposed that one probable explanation is that the physical therapists made subjective predictions about patients’ capacity for progress and that their actions reflect and convey these negative expectations [19]. Such ageist behaviours and thoughts towards older adults might lead to less use of DT in treating older people. A more recent study by Mannheim et al. [9] showed that healthcare professionals often have negative and ageist attitudes toward using DT with older adults and their technological abilities, thus raising the question of how DT is used in healthcare and raising the suspicion that ageism might lead to not providing DT-based treatment to older adults [9].

Therefore, negative attitudes towards the use of DT in healthcare for older adults are embedded in the existence of more general ageism in healthcare, for which different characteristics, social relationships and professional factors might be influential. First, there is evidence that more working experience, particularly experience working with older adults in one’s private life or work environment, is related to a less ageist attitude of healthcare professionals towards older adults [20,21,22]. In contradiction to these findings, other studies report that more exposure to older adults in the healthcare system also leads to more stereotypical beliefs about them, which the studies explained by the fact that most older adults encountered by healthcare professionals indeed have physical or mental conditions, leading to more dependency [23,24]. This contradiction in evidence is also found when looking at the personal characteristics of age and gender. While some studies have found that younger healthcare professionals tend to have more ageist attitudes towards older adults [20,22,25,26], others have concluded that older healthcare professionals may have a more ageist attitude towards older adults [27]. Additional studies found no correlation between the age of healthcare professionals and their attitudes towards older adults [28]. Regarding gender, studies have also shown mixed results, with some pointing out that men tend to have more ageist attitudes towards older patients [22,27,29,30]. In contrast, others found that women have more ageist attitudes towards older adults [21,26]. However, several studies found no significant relationship between gender and ageist attitudes towards older adults [20,28].

The studies above have found that healthcare professionals’ attitudes towards older adults may be influenced, to a certain degree, by their characteristics such as age, gender, experience, education level, and interest. However, no consistent explanation of the relationship between these characteristics and attitudes towards older adults has been found. Additionally, the widespread manifestations of ageism in healthcare against older persons in general [7] raises concerns about potential similar effects when using DT-based treatments with older patients [8].

Notably, there is little existing research focusing on ageism in allied healthcare professionals, and currently there are no studies investigating DT-based discrimination. Specifically, this study focuses on physiotherapists, as they are a big group within allied healthcare professionals. More so, using DT has a potential role in supporting self-management in people with musculoskeletal disorders [31]. This relates to disorders usually treated by a physiotherapist and is often seen in older patients for whom self-management means a more independent life. Therefore, this study’s research question is to investigate whether ageism and negative attitudes towards older adults’ abilities to use DT may influence the decisions of physiotherapists to use healthcare-related DT with older adults. Furthermore, this study investigates additional personal and social characteristics that may influence their attitudes toward using DT with older adults, such as gender, age, experience, and general attitudes toward using DT in care. Our primary hypothesis was that higher levels of ageism and negative attitudes of the physiotherapists towards older adults’ abilities to use DT would be associated with discriminatory behaviour related to offering DT-based treatments. As the literature on factors that might mitigate this association is inconclusive, we applied an exploratory approach. However, based on the literature on ageism, we predicted that younger professionals with less experience working with older persons would present higher levels of ageism and negative attitudes. 

## 2. Methods

### 2.1. Study Design

A cross-sectional research design was applied using an online questionnaire.

### 2.2. Participants 

Certified physiotherapists working in Luxembourg were recruited via an email invitation which was sent out by the Luxembourgish Physiotherapy association (ALK), and by a convenience sample through the authors’ professional networks and social media. The participants had to meet multiple inclusion criteria: specifically, being a certified physiotherapist, having work experience, actively working in Luxembourg, and working with adults older than 50. Physiotherapists from all settings (e.g., private setting, hospital setting) were eligible to participate.

An a priori statistical power analysis for logistic regression was performed to determine the needed sample size using G*Power 3.1.9.4. (Heinrich Heine University Düsseldorf, Düsseldorf, Germany), according to recommendations by Faul et al. [32]. A minimum sample size of 77 was calculated using the following assumptions: α of 0.05, assumed power of 90%, explained variance of our primary independent variable with other independent variables of 0.1 and the default odds ratio of 2.33.

### 2.3. Tools and Measurements

#### 2.3.1. Attitude towards Technology Use in the Work Environment 

In order to measure the attitudes of the participants towards the use of DT in their work environment as physiotherapists, we used 11 items modified from the Unified Theory of Acceptance and Use of Technology (UTAUT) model [33] and the extended Technology Acceptance Model (TAM2) [34]. The scale consists of 3 factors influencing the attitude towards using technology in the work environment: ‘Performance Expectancy’ (e.g., I think using DT makes my treatment more interesting for my patients), ‘Effort Expectancy’ (e.g., Working with DT is hard for me to understand), and ‘Social Influence’ (e.g., I think my colleagues will have a negative opinion about me when I use DT in my treatments). In addition, items related to the experience of using DT (e.g., I regularly use DT in my daily life.) were added to the scale, as the TAM2 suggests that the experience in using technology influences the attitude towards using it [34]. Participants were asked to rate the extent to which they agree or disagree with the statements on a Likert-type scale, ranging from 1 (totally disagree) to 6 (totally agree). To obtain the total score of the scale, six reversed items were recoded; then, the scores of all items were summed up, giving a score range of 11–66. A higher score is representative of a more positive attitude towards the use of DT at work. Venkatesh et al. [34] previously reported Cronbach’s α coefficients of above 0.8 in all cases for the TAM2 questionnaire. 

#### 2.3.2. Attitudes towards Older Adults Using Technology 

To measure participants’ DT-based ageism, we used the Attitudes Towards Older Adults Using Technology scale (ATOAUT-11) [9,35]. Items in this questionnaire are scored on a Likert-type scale from 1 (totally disagree) to 6 (totally agree). Three reversed items were recoded, and the scores of the 11 items were summed to obtain a score range from 11 to 66. A higher score represents a more negative attitude towards older adults using technology. A previous study with the scale reported a Cronbach α of 0.74 [35]. 

#### 2.3.3. Ageism

To measure participants’ ageism level, we used the 12-Item Expectations Regarding Aging Survey (ERA-12) [36]. The ERA-12 assesses the ‘stereotype’ dimension of ageism and is the ageism scale containing the highest psychometric properties [37]. Items were ranked on an ordinal scale from 1 to 4 (1, definitely true; 2, somewhat true; 3, somewhat false; 4, definitely false). For the combined score, 12 was subtracted from the integer of all scores, then multiplied by 25 and divided by 9 to obtain a score ranging from 0 to 100. A lower score indicates more negative expectations about ageing and is related to a higher level of ageism. This calculation and interpretation of the result is carried out according to Sarkisian et al. [36]. Previous studies reported a Cronbach’s α coefficient of 0.7–0.89 [36,38].

#### 2.3.4. Use of DT in Treatment

Additionally, participants were asked several questions on the actual use of DT in their treatment. Specifically, if they offer DT-based treatment to their patients and if they feel more comfortable/confident offering DT-based treatment to younger patients (under 50 years old) compared to older patients (over 50 years old). Responses for these questions were 0—never, 1—sometimes, 2—most of the time, or 3—always. Finally, participants were asked a dichotomous yes or no question: “In the past, I have NOT offered DT-based treatment to a patient because they were old/because of their age?”. Those who responded “yes” were also asked to indicate the reason for not offering DT-based treatment to older persons from the following options (participants could mark more than one category): I don’t think it is useful for them; I don’t have time or patience to explain it to them; I don’t think they will understand it; I don’t think they will like it/are interested in it.

#### 2.3.5. Socio-Demographic and Occupational Characteristics 

Participants answered questions about their socio-demographic and occupational characteristics, starting with their gender, age in years, work experience in years, setting that they work at, highest educational degree, and the percentage of patients above 50 that they treat. We specifically asked about patients over 50, as it was previously found that physiotherapists might consider people over 50 as less able to use digital technology [9].

### 2.4. Procedure

The data were collected between December 2021 and October 2022. Participants received an invitation link leading to the online questionnaire programmed on Google Forms. The questionnaire was initially in English, as most existing scales used in this study are available in English. Following this, the whole questionnaire was forward-translated to German and French, according to the recommendations of Koller et al. [39], by two independent evaluators; then, it was back-translated by two additional evaluators, all proficient in English, French, and German, so that participants could choose their preferred language. Pressing on the invitation link directed participants to the full information letter and the consent form, with which they had to agree to continue to the following questions. This information letter also included a definition of DT-based treatment: “digital health interventions to aid with home-based or in-clinic exercises. Examples of these interventions are web pages, mobile applications, video games that track movements, or wearables that contain, among other things, exercise videos, exercise pictures, reminders, or the ability to measure and record the patient’s outcomes”. To ensure that the participants met the inclusion and exclusion criteria, they had to answer three ‘kick-out-questions’. A ‘NO’ answer to any of these three questions meant that the participant did not meet the inclusion criteria for this study, and the questionnaire was ended at that point.

Following this, participants answered questions about their socio-demographic and occupational characteristics. Afterwards, participants filled in the questionnaire about the attitudes towards using technology in the work environment, the ATOAUT-11, the ERA-12, questions about the actual use of DT-based treatment, and the direct yes or no question on if they had ever in the past not offered a DT-based treatment to a patient because of their age.

### 2.5. Data Analysis

The data analysis was performed using IBM SPSS Statistics version 28 (IBM, Armonk, NY, USA) for Windows. Characteristics were analysed using descriptive statistics. Cronbach’s α coefficient for all scales used in the study was calculated to determine internal consistency.

A correlation matrix was created for insight into the relation between all variables. Stepwise multiple logistic regression was used to test our main hypothesis, with the dichotomous variable of not offering DT-based treatment in the past because of one’s age as the dependent variable (yes = 1, no = 0). We applied bootstrapping with n = 1000 samples. In the initial step, we added our primary independent variable, ATOAUT-11. In the second step, general ageism was added (ERA-12), and in the final step, additional variables were added; namely, attitudes towards using DT in the work environment, gender, age, work experience, and the percentage of patients above 50. *p*-values < 0.05 were considered statistically significant for all analyses.

## 3. Results

Initially, 85 people responded to the questionnaire. In total, seven participants were excluded for not meeting the inclusion criteria. Two participants were excluded because they did not yet have a degree in physiotherapy, two because they did not work as physiotherapists in Luxemburg and three because they indicated they did not work with patients above 50. The final sample for analysis was thus n = 78. The mean age of the study population was 38.74 (SD = 9.78, range 23–64). The sample was equally divided between male and female (51.3% female). Participants had an average of 14.1 years of work experience (SD = 9.9), ranging from less than one year of work experience to 40 years of work experience. A total of 59% of the participants worked with 50% or more patients older than 50 years, of whom five worked exclusively with adults over the age of 50. Most participants preferred to complete the questionnaire in French (64.1%), which is consistent with the fact that most participants studied in Belgium (57.7%) or France (5.2%), where French is one of the official languages. A total of 32.1% completed the questionnaire in German (20.5% studied in Germany), and 3.8% in English. More than half of the participants had a master’s degree (61.6%), and 71.8% worked in a private practice.

Considering the participants’ actual use of DT-based treatment, most participants (61.5%) stated that they offered DT-based treatment to their patients sometimes or most of the time, whereas 38.5% never offer it to their patients. A total of 10.3% reported that they would always feel more comfortable offering DT-based treatment to younger patients than older patients, while more than half of the participants (62.8%) indicated feeling more comfortable at least sometimes, and 26.9% indicated no difference. 

Twenty-four participants (30.8%) reported that they had, in the past, not offered DT-based treatment to a patient because of the patient’s older age. As the main reasons for not offering DT-based treatment in the past, the participants indicated (participants could select more than one response): that they do not think older patients see a benefit in using it (62.5%); that older patients do not understand it (41.7%); that older patients do not like it/are not interested in it (41.7%); or that they (the physiotherapists) do not have the time or patience to explain to the older patient how to work with it (41.7%). 

Table 1 presents the mean scores, Standard Deviations (SD), and correlations between the different scales and main variables used in this study. Examining skewness and kurtosis values revealed no extreme violations of normality. The mean score of the ATOAUT-11 scale was 44.29 (of max 66 of the scale, SD = 9.83), demonstrating relatively high (negative) attitudes towards older adults’ abilities to use DT. Higher scores were significantly correlated with feeling more comfortable offering DT-based treatment to younger patients and not offering DT-based treatment to older patients in the past. Additionally, having a higher ATOAUT score showed a statistically significant correlation with less positive attitudes towards using technology in the work environment. The Expectations Regarding Aging (ERA-12) scale was not significantly correlated with ATOAUT-11. However, lower (negative) ERA was correlated with the older age of the physiotherapists, more work experience and working with a higher percentage of patients above 50.

More so, higher (positive) attitudes towards technology use in the work environment were correlated with the younger age of the physiotherapists, the actual use of DT-based treatment with patients, and feeling more comfortable with offering DT-based treatment to younger patients. No correlations were identified with gender.

Cronbach’s alpha for all scales were respectively high, and comparable with previous literature, suggesting a good reliability of the scales used (0.819 for the Attitudes Towards Technology Use in the Work Environment, 0.845 for the ATOAUT scale, and 0.791 for the ERA-12 scale).

To test our main hypothesis, we performed a stepwise logistic regression (see Table 2). The main assumptions for logistic regression (independence of errors, linearity in the logit for continuous variables, absence of multicollinearity, and absence of influential outliers) were not violated. Adding the ATOAUT-11 in the first step revealed a significant model (χ2(1) = 5.022, *p* = 0.025, Nagelkerke R^2^ = 0.088). The odds ratio of ATOAUT-11 was 1.061 (*p* = 0.032). Adding general ageism (ERA-12) in the second step did not add significant variance. Adding all other characteristic variables in the last step increased the explained variance (Nagelkerke R^2^ = 0.226). However, the model did not reach statistical significance (*p* = 0.058). Examining the odds ratio of ATOAUT in the third step showed that for an increase of one unit of ATOAUT (more negative attitudes) when all other variables are fixed, the probability of not offering a DT-based treatment in the past increases by 1.082 (*p* = 0.013). Thus, for an increase of one standard deviation in ATOAUT (SD = 9.834), the odds of not offering DT-based treatment in the past increased by 2.17. All other variables did not show a significant odds ratio, except for general attitudes towards using DT in the work environment, for which the odds ratio for an increase in (positive) attitudes was 1.084 (*p* = 0.024).

## 4. Discussion

This study aimed to investigate potential DT-related ageism in physiotherapy and to find additional personal and social characteristics possibly influencing physiotherapists’ attitudes towards using DT with older adults. To the best of our knowledge, this is the first study to demonstrate the influence of ageism, particularly technology-specific ageism (measured by the ATOAUT-11), that may lead to discriminatory decisions of physiotherapists to offer or not offer DT-based treatment. Our main result showed that participants with more negative ATOAUT-11 scores (1 SD higher) were two times more likely not to have offered DT-based treatment in the past due to the patient’s older age. This finding shows that the attitude towards older adults and technology may influence physiotherapists’ actual use of DT-based treatments for older adults.

Our results were confirmed by the observation that not offering a DT-based treatment to an older patient in the past was correlated with feeling more comfortable offering DT-based treatment to younger patients compared to older patients. In addition, the results showed that the actual use of DT-based treatment in general was related to a more positive attitude towards using DT in the work environment. These correlations confirm that the physiotherapists’ attitudes towards using DT at their workplace or with older patients can influence whether they use it with their older patients or not.

Earlier studies revealed factors that could potentially influence the physiotherapists’ attitudes towards the use of DT with older persons, such as age, gender, work experience, and experience of working with older adults [20,21,22,23,24,25,26,27,28,29,30]. In contrast, in our study, only positive attitudes towards the use of DT at work were associated to not offering DT-based treatment in the past. While the latter finding might seem contradictory, it may be explained by the correlations between positive attitudes toward using DT at work and a higher use of DT-based treatments. Thus, physiotherapists who use more DT in their treatments might be more prone to not offering them when potentially having higher levels of DT-specific ageism.

Notably, we did not find that higher levels of general ageism (as measured by the Expectations Regarding Aging scale—ERA-12) were associated with discriminatory behaviour related to offering DT-based treatments. Interestingly, negative expectations regarding ageing were correlated with the older age of the physiotherapist, having more work experience and a higher percentage of patients over 50. This finding may seem somewhat surprising considering Contact Theory [40], which would posit that higher contact should reduce intergroup prejudice, and prior research that suggests that more exposure to older persons leads to reduced ageism among healthcare professionals [20,21,22]. Nevertheless, this finding was also found in previous literature and helps to underline that perhaps higher exposure to older adults in healthcare who might confirm older age stereotypes of frailty and dependency may lead to more negative beliefs about ageing and older adults [23,24,41]. Besides the conflicting results about the influencing characteristics, previous studies have identified manifestations of ageism among physiotherapists and healthcare professionals. Hence, the existence of some level of ageism in this study group is not surprising [9,14,15,16,42,43].

Additionally, we did not find a direct association between general ageism and ATOAUT. A previous study has shown that this correlation could be moderated by implicit processes and low awareness [8]. Therefore, it should not be assumed that ageism and DT-specific ageism among healthcare professionals are intentional, as this may indeed be without awareness or intention to do harm [44].

The inconsistency of the findings in previous literature and the inconclusiveness of the findings in this study do not allow us to infer conclusions about which specific characteristics of physiotherapists would lead to more (or less) discrimination towards older patients. More research is warranted to unravel the potential influence of these characteristics, using more prominent and representative samples of various healthcare professionals in which additional intersections and more complex and parsimonious models can be tested. 

Our main observation, i.e., that having a more negative attitude towards older adults using DT was associated with not offering DT-based treatment to older adults implies that there is a need to change physiotherapists’ attitudes to ensure that older patients receive the best possible care. Suggestions to achieve this can be to raise awareness during the education of physiotherapists, which means that during their training, physiotherapy students should be shown the possibilities of using DT in treatment and receive training on how it can be used with different patient groups. In addition, the care of older adults should, in general, be addressed during the education of physiotherapy students, as it is a substantial and increasing part of the patients they will care for in their future profession. Another suggestion is to offer training for experienced physiotherapists, as many DT-based treatments are relatively new and not all physiotherapists had the chance to be sufficiently trained on the use of DT during their education. This means that many physiotherapists might need to learn what possibilities the use of DT would give them, in general, and specifically, in the treatment of older adults. One big opportunity in using DT in the treatment of older adults would be to gamify exercises by using interactive digital tools. This can lead to improved adherence to the prescribed rehabilitation routines [45]. Additionally, the use of remote DT can help to monitor a patient’s activity throughout the day without being physically present. Wearables for monitoring, for example, could give the physiotherapist an objective idea on the activity level of the patient and would help to adapt the rehabilitation plan accordingly [46].

Finally, following the correlations found with general ageism and several characteristics, we suggest raising awareness about ageism’s prevalence and potential harmful ramifications in healthcare. Physiotherapists, as well as other healthcare professionals, should learn to reflect on their behaviour and be aware of the prejudices they have towards their patients. This can be done during courses, campaigns, or symposiums about the topic. 

### Strengths and Limitations

The outcome of the descriptive statistics shows some strengths of this study. For instance, there is a balance between genders and working experience in the study population. The age range from 23 to 64 years also suggests that various ages and experiences are represented in our sample.

However, this study has some limitations. First, the sample size of 78 participants, while meeting minimal requirements according to our a priori power analysis, may have needed to be bigger to achieve the required statistical power of 90%, which is needed to detect an existent effect. This study was also conducted in Luxemburg, a relatively small country in Europe. Thus, future research using a larger sample size in additional countries would also allow the use of mediators in a more complex analysis, generalise these findings, and help to find more conclusive results regarding the specific characteristics influencing discriminatory attitudes among physiotherapists. Having said that, most of our sample studied physiotherapy outside Luxemburg (mainly, Belgium, France, and Germany), thus perhaps representing approaches from various European countries. Nevertheless, different studies from different countries (also outside Europe) may provide the opportunity to consider the variation of using DT between countries, which might be relevant for the use of DT-based physiotherapeutic treatment with older adults. Finding out about these differences and the reasons why they exist would lead to a better understanding of the relative importance of ageism in the use of DT in therapeutic settings. 

Another limitation might have been the way we formulated the question of if, in the past, the participant did not offer DT-based treatment to a patient because of their older age. The social desirability inferred from this question may have suggested that answering ‘yes’ would reflect a negative attribute to the participants and might have led more people to answer ‘no’ even though, in reality, they may not have offered DT-based treatment. Nonetheless, 31% of the participants still answered this question with ‘yes’. Using a more subtle or indirect way of measuring use behaviour of DT-based treatment in the future may assist in identifying more substantial effects. Finally, this study was restricted to physiotherapists. Although they represent a significant group of allied health professionals, they do not represent other professions, such as speech therapists, occupational therapists, and dieticians. It is also advised to broaden the group of allied health professions in future research.

## 5. Conclusions

Ageism in general is prevalent in healthcare, and as also identified in this study can be influenced by specific characteristics such as age, experience, and exposure to older patients. Most importantly, this study among physiotherapists in Luxembourg demonstrates for the first time that DT-based ageism can lead to discrimination and deprive older persons of receiving optimal physiotherapy treatment. Knowing this, it is important to ensure that the education of physiotherapists and allied health care professionals in general is managed in a way that guarantees they are aware of this phenomenon. This would lead to changes in their thinking and improve care trajectories for older patients in the future. In addition, it is also important to offer experienced physiotherapists the opportunity to train themselves in the use of emerging technologies, so they are knowledgeable on how to use them with patients of any age. More research is needed to unravel the phenomenon of ageism in treatment among other allied healthcare professions, identify other characteristics that influence the use of technology with older persons, and test interventions that can reduce ageism and lead to increasing equality in DT-based healthcare. 

## Figures and Tables

**Table 1 healthcare-11-02672-t001:** Mean scores, standard deviations and Pearson correlations between main variables (n = 78).

	Mean (SD)	1	2	3	4	5	6	7	8	9
1. ATOAUT	44.29 (9.83)	-								
2. ERA	55.52 (14.82)	−0.115	-							
3. Attitudes towards use of DT in the work environment	44.05 (9.03)	−0.238 *	0.038	-						
4. Age (years)	38.74 (9.78)	0.086	−0.252 *	−0.269 *	-					
5. Gender ^a^	0.51 (0.50)	0.121	0.080	−0.186	−0.065	-				
6. Percentage of patients above 50	53.33 (24.64)	0.082	−0.278 *	−0.046	0.239 *	−0.035	-			
7. Work experience (years)	14.10 (9.90)	0.074	−0.278 *	−0.271 *	0.958 ***	−0.088	0.245 *	-		
8. Use of DT-based treatment ^b^	0.67 (0.57)	−0.100	−0.023	0.643 ***	−0.208	−0.120	−0.113	−0.263 *	-	
9. More comfortable offering DT-based treatment to younger patients ^b^	1.27 (0.98)	0.237 *	−0.144	0.314 **	−0.043	0.033	0.059	−0.026	0.487 ***	-
10. Did not offer DT-based treatment in the past because of patient’s older age ^c^	0.31 (0.46)	0.250 *	−0.066	0.194	−0.094	0.038	0.045	−0.053	0.292 **	0.531 ***

* Correlation is significant at the 0.05 level (2-tailed). ** Correlation is significant at the 0.01 level (2-tailed). *** Correlation is significant at the 0.01 (2-tailed). ATOAUT: Attitudes Towards Older Adults Using Technology, ERA: Expectations Regarding Aging, DT: Digital Technology. ^a^ Coded as male = 0, female = 1. ^b^ Coded as 0 = never; 1 = sometimes; 2 = most of the time; 3 = always. ^c^ Coded as no = 0, yes = 1.

**Table 2 healthcare-11-02672-t002:** Logistic regression of not offering DT-based treatment (n = 78).

								95% C.I. for Exp(B)	
Step		B	S.E.	Wald	df	Sig.	Exp(B)	Lower	Upper
1	ATOAUT	0.059	0.027	4.625	1	0.032	1.061	1.005	1.119
	Constant	−3.477	1.291	7.249	1	0.007	0.031		
2	ATOAUT	0.058	0.027	4.430	1	0.035	1.059	1.004	1.118
	ERA	−0.006	0.018	0.123	1	0.726	0.994	0.960	1.029
	Constant	−3.088	1.690	3.340	1	0.068	0.046		
3	ATOAUT	0.079	0.032	6.206	1	0.013	1.082	1.017	1.152
	ERA	−0.007	0.020	0.115	1	0.735	0.993	0.954	1.034
	Attitudes towards use of DT in the work environment	0.081	0.036	5.093	1	0.024	1.084	1.011	1.163
	Gender	0.318	0.584	0.297	1	0.586	1.375	0.437	4.322
	Age (years)	−0.151	0.105	2.046	1	0.153	0.860	0.700	1.057
	Work experience (years)	0.140	0.105	1.781	1	0.182	1.150	0.937	1.413
	Percentage of patients above 50	0.124	0.595	0.044	1	0.835	1.132	0.353	3.634
	Constant	−4.046	3.783	1.144	1	0.285	0.017		

ATOAUT: Attitudes Towards Older Adults Using Technology; ERA: Expectations Regarding Aging; gender coded as male = 0, female = 1; percentage of patients above 50 coded as below 50% = 0, 50% and above = 1.

## Data Availability

The corresponding author may be contacted for further inquiries regarding the measurements used and the data sets.

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
