# Peer review of "The Association of Technology-Based Ageism with Using Digital Technology in Physical Therapy for Older Persons"

_healthcare, 2023, doi:10.3390/healthcare11192672_

Round 1

Reviewer 1 Report

Dear authors,

Thank you for the opportunity to review your interesting manuscript. I will give my feedback following the structure of the manuscript. 

1.Title and abstract

The title is informative and the abstract provides a summary of the manuscript's major aspects. 

2.Introduction

The background chapter is clear, well developed, well structured and well referenced. However I am missing information about the importance of DT for physiotherapists. In my opinion, the authors should explain this issue in the introduction section. 

3.Methods

Participants

This section contains information on the design of the study and should go in a previously separate section (study design).

Regarding the participants, I lack information on the inclusion and exclusion criteria and also information on the type of sampling (convenience sampling?) . In my opinion authors should include this information in the participants section. 

Also, there is no information in the manuscript about the setting (the authors only explain that participants are physiotherapists working in Luxemburg with adults over 50). But, do they work in a primary care setting? or a hospital setting?... ). Thus, the authors could add more information about the setting in which the participants work in this section. 

Tools and Measurements

In my opinion this section is very clear. However I am missing information on the socio-demographic and occupational characteristics of the participants. If authors have collected this information, they should explain it in this section. 

Procedure

I would like to congratulate the authors for these sections, however I find some information missing:

-Who sent the questionnaires to the participants? 

-The authors explain how the questionnaire was sent and we know beforehand that they need a sample of 77 participants. However, we do not know how many professionals have received the questionnaire and have the opportunity to participate in the study. Could the authors clarify this point? 

-Are there no validated translations of the scales used? In my opinion the authors should clarified this point.

-It would be better to move the information on the exclusion criteria for participants to the participants  section. 

-In the procedures section, the authors indicate that they have collected information on gender, age, work experience... This information must also be explained in the tools section.

-The authors have added the information about the ethics committee in this section. In my opinion all the information about ethical considerations should be  in a separate section: ethical considerations, where the authors should explain all ethical considerations, for example how they maintained tha anonymity of the participants. 

Data Analysis

In my opinion this section is very clear. Nothing to add.

4. Results 

I would like to congratulate the authors for this section. In my opinion it is well-written and clear.  I just have one suggestion about the information regarding why it doesn't offer DT-based treatment to the elderly (line 238 to line 242). Are these results from the explained question from line 169 to line 171? "Those who answered 169 'yes' were also asked to indicate the reason for not offering DT-based treatment to the elderly." I had understood this question as qualitative and in the results section it is presented as quantitative. Could the authors clarify this point?

5.Discussion

I would like to congratulate the authors for this section. It’s very clear and discusses the most relevant results of their study. I only have some suggestions: 

-In line 297, the authors say that previous studies have indicated some factors that may influence the physiotherapist’s attitudes towards the use of DT with older persons, but they do not give us any reference of these studies. Could they add the references of these previous studies? 

-In the last paragraph of this section the authors give some recommendations for increasing DT-based treatment in the elderly, but do not discuss why this treatment is so important. In my opinion, reinforcing this idea could improve this last part of the discussion. 

-The results of this study are from professionals in a European country. In my opinion, it would be interesting to discuss whether DT-based physiotherapist treatment in the elderly is common in other European countries.

Strengths and Limitations

In my opinion, this section is very clear. Nothing to add. 

5.Conclusions

In my opinion, this section could be improved. After reading the results, I think the authors could conclude more ideas than they have presented in this section. In addition, in this section they could add some recommendations or suggestions to improve DT based treatment for the elderly. I encourage the authors to rethink this section. 

Author Contributions: 

-How can readers know who each author is? (XX, XY, ZZ). 

Funding/Institutional Review Board Statement/Informed Consent Statement:

The authors need to review the information given in these sections and only state the information referred to their study.

Author Response

Response to reviewers healthcare-2601932 entitled " The Association of Technology-Based Ageism with Using Digital Technology in Physical Therapy for Older Persons"

Dear Editor and reviewers,

We wish to thank you for reviewing our paper and providing us with useful and constructive feedback. We are very happy that the reviewers found the topic of our paper “interesting and current issue”. Therefore we were enthusiastic and motivated to make the needed improvements in order to communicate our findings to the readership of the Healthcare Journal.

Please find our response to the reviewers’ comments below. We numbered each comment of the reviewers and provided detailed answers as to the changes and improvements we made to address these in the manuscript.

For your convenience, we marked all changes in the manuscript in red.

We hope you find this revision has improved the manuscript and further consider it for publication. 

Kind regards, on behalf of the authors,

Reviewer: 1

Dear authors, thank you for the opportunity to review your interesting manuscript. I will give my feedback following the structure of the manuscript.

  1. Title and abstract: The title is informative and the abstract provides a summary of the manuscript's major aspects. 

      Thank you for this positive feedback.

  1. Introduction: The background chapter is clear, well developed, well structured and well referenced. However I am missing information about the importance of DT for physiotherapists. In my opinion, the authors should explain this issue in the introduction section. 

We agree with this part being less specifically addressed in our introduction. In general, we noticed that several studies describe the potential of applying digital technology in healthcare and treatment in general (1-2, 8). We added more specific information about DT and physiotherapy in the last paragraph of our introduction referring to a recent review by Kelly et al., and additional references in the discussion (see added reference below). However, there is a limited amount of information available regarding the intersection of Digital Technology and ageism within the field of physiotherapy, for which we provided 2 references (9,19). Our paper addresses this by offering insights into the intersection of DT and ageism in physiotherapy, which sets it apart from previous research in the field.

Kelly, M.; Fullen, B.M.; Martin, D.; Bradley, C.; McVeigh, G.J. eHealth interventions to support self-management: Perceptions and experiences of people with musculoskeletal disorders and physiotherapists - ‘eHealth: It’s TIME’: A Scoping Review. Physical Therapy 2022, 102(4), pzab307 doi:10.1080/09593985.2022.2151334

Alfieri, F. M.; da Silva Dias, C.; de Oliveira, N. C.; & Battistella, L. R. Gamification in musculoskeletal rehabilitation 2022. Current Reviews in Musculoskeletal Medicine, 15, 629-636.

Hannan, A. L.; Harders, M. P.; Hing, W., Climstein, M.; Coombes, J. S.; & Furness, J. Impact of wearable physical activity monitoring devices with exercise prescription or advice in the maintenance phase of cardiac rehabilitation: systematic review and meta-analysis. BMC Sports Science, Medicine and Rehabilitation 2019, 11(1), 1-21.

  1. Methods-Participants: This section contains information on the design of the study and should go in a previously separate section (study design).

Thank you for pointing this out. Following your advice, we added a sub-section about study design in the method section.

  1. Methods-Participants: Regarding the participants, I lack information on the inclusion and exclusion criteria and also information on the type of sampling (convenience sampling?). In my opinion authors should include this information in the participants section. 

Thank you for this comment. We have now moved the detailed part of the inclusion and exclusion criteria from the procedure section to the participants’ section and have given more detail about the type of sampling that we did. We stated the following in the manuscript: ‘Certified physiotherapists working in Luxembourg were recruited via an email invitation which was sent out by the Luxembourgish Physiotherapy association (ALK), and by a convenience sample through the authors’ professional networks and social media. The participants had to meet multiple inclusion criteria. Specifically, being a certified physiotherapist, having work experience, actively working in Luxembourg, and working with adults older than 50. Physiotherapists from all settings (e.g., private setting, hospital setting) were eligible to participate.’

  1. Methods-Participants: Also, there is no information in the manuscript about the setting (the authors only explain that participants are physiotherapists working in Luxemburg with adults over 50). But, do they work in a primary care setting? or a hospital setting?... ). Thus, the authors could add more information about the setting in which the participants work in this

Thank you for pointing out this unclarity. In our study we have included participants from all care settings. As also described in comment 4, we have clarified this now in the manuscript by stating the following under Participants: ‘Physiotherapists from all settings (e.g., private setting, hospital setting) were eligible to participate’.

  1. Methods-Tools and Measurements: In my opinion this section is very clear. However I am missing information on the socio-demographic and occupational characteristics of the participants. If authors have collected this information, they should explain it in this section. 

Thank you for the positive feedback on the clarity of this section. Considering your concern about the socio-demographic and occupational characteristics and in accordance with your comment number 12, we added a subsection about these characteristics at the end of the ‘Tools and Measurements’ section, where we give detail on which characteristics we collected. We subsequently listed these sample specific details about these characteristics in the first paragraph of the results section.

  1. Methods-Procedure: I would like to congratulate the authors for these sections, however I find some information missing:

Thank you for your positive feedback and pointing out the missing information.

  1. Methods-Procedure: Who sent the questionnaires to the participants?

Thank you for pointing out this unclarity. We added more detail about this topic. Also, in accordance with your comment 5, we added the details in the participants sub-section to make it more clear how the participants were sampled: “Certified physiotherapists working in Luxembourg were recruited via an email invitation which was sent out by the Luxembourgish Physiotherapy association (ALK), and by a convenience sample through the authors’ professional networks and social media.”

  1. Methods-Procedure: The authors explain how the questionnaire was sent and we know beforehand that they need a sample of 77 participants. However, we do not know how many professionals have received the questionnaire and have the opportunity to participate in the study. Could the authors clarify this point? 

As the emails were sent out by the Luxembourgish Physiotherapy Association to their members and there was additional convenience sampling through our networks and social media, we can’t retrace how many people were reached through the sampling process and therefore we could not give detailed information about this in the paper.

  1. Methods-Procedure: Are there no validated translations of the scales used? In my opinion the authors should clarified this point.

We did not use any validated translations of the scales used as they were not available in the languages we needed. Therefore, we used forward-backward translation procedure that is advised for translation of validated questionnaires (see added reference below). We mentioned this in the procedure section, and indicated that the: ‘whole questionnaire was forward translated to German and French, according to the recommendations of Koller et al. [38], by two independent evaluators and then back-translated by two additional evaluators, all proficient in English, French and German, so that participants could choose their preferred language’.  

Koller, M.; Aaronson, N. K.; Blazeby, J.; Bottomley, A.; Dewolf, L.; Fayers, P.; Johnson, C.; Ramage, J.; Scott, N.; West, Karen. Translation procedures for standardised quality of life questionnaires: The European Organisation for Research and Treatment of Cancer (EORTC) approach. European Journal of Cancer 2007,43, 1810-1820.

  1. Methods-Procedure: It would be better to move the information on the exclusion criteria for participants to the participants’ section. 

Thank you for pointing this out. We moved the detailed information about the in- and exclusion criteria to the participants section.

  1. Methods-Procedure: In the procedures section, the authors indicate that they have collected information on gender, age, work experience... This information must also be explained in the tools section.

See our response on comment 6.

  1. Methods-Procedure: The authors have added the information about the ethics committee in this section. In my opinion all the information about ethical considerations should be in a separate section: ethical considerations, where the authors should explain all ethical considerations, for example how they maintained the anonymity of the participants.

Thank you for your suggestion. Following your advice, we have moved the information about ethics to the appropriate section according to the journal format, and added more information on anonymity and consent in the relevant section.

  1. Data Analysis: In my opinion this section is very clear. Nothing to add.

Thank you very much for this positive feedback.

  1. Results:I would like to congratulate the authors for this section. In my opinion it is well-written and clear.  I just have one suggestion about the information regarding why it doesn't offer DT-based treatment to the elderly (line 238 to line 242). Are these results from the explained question from line 169 to line 171? "Those who answered 169 'yes' were also asked to indicate the reason for not offering DT-based treatment to the elderly." I had understood this question as qualitative and in the results section it is presented as quantitative. Could the authors clarify this point?

Thank you for your positive feedback about the result section. Also, thank you for pointing out the confusion. We used a yes or no question to ask if or if not the participant did not offer DT-based treatment to a patient because of their age. Following, we asked in a second question – when they answered yes - about the reasons why. This question was a multiple response question with a closed list of possible responses (and participants could select more than one category). This is clarified now in the ‘Tools’ section.  

  1. Discussion: I would like to congratulate the authors for this section. It’s very clear and discusses the most relevant results of their study. I only have some suggestions: Discussion: In line 297, the authors say that previous studies have indicated some factors that may influence the physiotherapist’s attitudes towards the use of DT with older persons, but they do not give us any reference of these studies. Could they add the references of these previous studies? 

Thank you for pointing out the lack of references. We added the references to the previous studies [10-20] and also rephrased the sentence according to comment 26 of reviewer 2.

  1. Discussion: In the last paragraph of this section the authors give some recommendations for increasing DT-based treatment in the elderly, but do not discuss why this treatment is so important. In my opinion, reinforcing this idea could improve this last part of the discussion. 

We agree with this comment and added suggestions of how the use of DT can benefit the physiotherapist to make the treatment better for the patient in question.

  1. Discussion: The results of this study are from professionals in a European country. In my opinion, it would be interesting to discuss whether DT-based physiotherapist treatment in the elderly is common in other European countries.

There are no specific papers on this topic. So taking this idea we think this could be a recommendation for future research. With additional thoughts of being able to compare between countries and use the findings to find reasons that could lead to not using DT in the treatment with older adults. We added these thoughts to the discussion and conclusion of our manuscript.

  1. Discussion-Strengths and Limitations: In my opinion, this section is very clear. Nothing to add. 

Thank you for this positive feedback. Notably, following comments from the other reviewers we revised the discussion.

  1. Conclusions: In my opinion, this section could be improved. After reading the results, I think the authors could conclude more ideas than they have presented in this section. In addition, in this section they could add some recommendations or suggestions to improve DT based treatment for the elderly. I encourage the authors to rethink this section. 

We agree with this comment and therefore incorporated more concrete ideas in the conclusion to make the conclusion more complete and practical. We especially focused on suggestions on how to reduce DT-based ageism amongst physiotherapists and the education of physiotherapists.

  1. Author Contributions:How can readers know who each author is? (XX, XY, ZZ). 

Thank you for noticing, we filled this online, but this was indeed not added to the version you received. We have added this section.

  1. Funding/Institutional Review Board Statement/Informed Consent Statement: The authors need to review the information given in these sections and only state the information referred to their study.

Thank you for noticing, we filled this online, but this was indeed not added to the version you received. We have added this section.

Reviewer 2 Report

Thank you for the opportunity to read the manuscript. 

This is a study that evaluates physiotherapists' ageism and its impact on technology-based treatments.

The article is very well-written and interesting. I have a few minor suggestions.

Lines 62-64: not clear. Please re-phrase.

Lines 96-98: not clear. Please re-phrase.

Lines 299-301: not clear. Please re-phrase.

Conclusion: Please elaborate a little about your findings and the implications.

Author Response

Response to reviewers - healthcare-2601932 entitled " The Association of Technology-Based Ageism with Using Digital Technology in Physical Therapy for Older Persons"

Dear Editor and reviewers,

We wish to thank you for reviewing our paper and providing us with useful and constructive feedback. We are very happy that the reviewers found the topic of our paper “interesting and current issue”. Therefore we were enthusiastic and motivated to make the needed improvements in order to communicate our findings to the readership of the Healthcare Journal.

Please find our response to the reviewers’ comments below. We numbered each comment of the reviewers and provided detailed answers as to the changes and improvements we made to address these in the manuscript.

For your convenience, we marked all changes in the manuscript in red.

We hope you find this revision has improved the manuscript and further consider it for publication. 

Kind regards, on behalf of the authors,

Reviewer: 2

23. Thank you for the opportunity to read the manuscript. This is a study that evaluates physiotherapists' ageism and its impact on technology-based treatments. The article is very well-written and interesting. I have a few minor suggestions.

Thank you for this positive feedback on our writing and for your interest in reading it. We tried to make changes according to your comments.

24. Lines 62-64: not clear. Please re-phrase.

Thank you for pointing this out we rephrased the sentence to make it clearer: ‘In the context of physiotherapy, Ambady et al. [19] discovered that non-verbal communication of physiotherapists during treatment - such as distancing themselves looking away from the patient - was associated with physical decline in the older patients, both in the short term and the long term.’

25. Lines 96-98: not clear. Please re-phrase.

Thank you for pointing this out. We rephrased the sentence to make it clearer: ‘Additionally, the widespread manifestations of ageism in healthcare for older persons in general [7], raises concerns about potential similar effects when using DT-based treatments with older patients [8].’

26. Lines 299-301: not clear. Please re-phrase.

Thank you for pointing this out we split the sentence into 2 sentences to make it clearer: ‘Earlier studies revealed factors that could potentially influence the physiotherapists’ attitudes towards the use of DT with older persons, such as age, gender, work experience, experience of working with older adults [20-30]. In contrast, in our study we only found that positive attitudes towards the use of DT at work was associated to not offering DT-based treatment in the past.’

27. Conclusion: Please elaborate a little about your findings and the implications.

See also comment 20 of reviewer 1. We incorporated more details in the conclusion to make the conclusion more complete. We especially focused on practical suggestions on how to reduce DT-based ageism amongst physiotherapists and education of physiotherapists.

Reviewer 3 Report

Review

The Association of Technology-Based Ageism with Using Digital Technology in Physical Therapy for Older Persons

The authors conduct a study in which they investigate possible ageism related to DT in physical therapy and find additional personal and social characteristics that possibly influence their attitudes toward wards using DT with older adults. From my point of view, I think it is an interesting and current issue due to the digital world in which we live and it is necessary to know if there are prejudices when using technology because it is important that the whole society can benefit from its use.

Despite this, I believe that the work has many weaknesses and could be improved.

Specific comments

List of authors and affiliation

The e-mail address of each author must be provided

Abstract

The abstract should be a single paragraph and should follow the style of structured abstracts, but without headings: 1) Background, 2) Methods,3) Results, 4) Conclusion.

Introduction

DT or DTs?

The study highlights that this issue may be further compounded by gender, as older women may be more likely to experience ageism in their interactions with DT.

Authors should give the reference of the study. Is it Choi et al.

Chang et al. found in their systematic review that 84.6% of the 56 studies they included showed that chronological age dictates which treatment or procedure will be chosen [16], meaning that a younger patient has a better outlook on receiving  the gold standard treatment.

Authors should place the reference after the author's name.. Chang et al. [16]

For example, the systematic review looks at patients awaiting breast cancer surgeries and found in multiple studies that older patients were much more likely to be rejected for treatment than younger patients with identical histories. Rejected from what treatment?

The authors relied on a previous study to calculate the sample size. Why these values? An a-priori statistical power analysis for logistic regression was done to determine the needed sample size using G*Power 3.1.9.4. (Heinrich Heine University Düsseldorf). A minimum sample size of 77 was calculated using the following assumptions: α of 0.05, assumed power of 90%, explained variance of our primary independent variable with other independent variables of 0.1 and the default odds ratio of 2.33.

Methodology

A section on the type of study is missing

In this section, I suggest to put also the information of the ethics committee.

Instrument

A Cronbach's α coefficient of .819 was found for the scale. Is there any reference data for this instrument?

Personally, in this methodology section I would only put the reference values of Conbrach's. The values obtained in this study I would put in the reference values of Conbrach's. I would put those obtained in this study in the results section.

Produce

How did you get in touch with the sample? They were sent an invitation by mail but how did you get the mails? Through professional associations...?

Analysis

Was the assumption of normality tested? Results are expressed as means and SD. These statistics are used for normal samples.

SPSS Statistics 28 (IBM US). Use the full reference

Suggestion: IBM SPSS Statistics for Windows, Version 28.0. Armonk, NY: IBM Corp.

A correlation matrix was created for insight into the relation between all variables. by which statistical test? Pearson, Sperman?

Were the applicability assumptions of logistic regression tested?

Results

The wording of the results should include the main results in terms of numbers (more next findings).

No table of descriptive statistics?

Line 237: not is write in italics

In the line 244: Standard Deviations (SD). This should appear the first time the abbreviation appears.

Line 247: compared to previous results with physiotherapists [8]. In the results section, results cannot be discussed with other studies.

Line 246: ATOAUT-11 scale was 44.29 (of max 66)?because no SD is given here?

Correlations with qualitative data?

+ Correlation is significant at the 0.1 level (2-tailed)?

However, the model was only marginally significant at a p=0.058 level. This is not significant. It should be < 0.05. The authors also put it in the analysis section: P-values <0.05 were considered statistically significant for all analyses

Table 1. Should be reorganized

In the tables, all abbreviations should be described in the table footnote.

Discussion

Could be improved

Conclusion

It should be specified that the findings are in terms of the sample studied.

Author Contributions: I think this section is not filled in.

Funding: I think this section is not filled in.

Informed Consent Statement: I think this section is not filled in.

Author Response

Response to reviewers - healthcare-2601932 entitled " The Association of Technology-Based Ageism with Using Digital Technology in Physical Therapy for Older Persons"

Dear Editor and reviewers,

We wish to thank you for reviewing our paper and providing us with useful and constructive feedback. We are very happy that the reviewers found the topic of our paper “interesting and current issue”. Therefore we were enthusiastic and motivated to make the needed improvements in order to communicate our findings to the readership of the Healthcare Journal.

Please find our response to the reviewers’ comments below. We numbered each comment of the reviewers and provided detailed answers as to the changes and improvements we made to address these in the manuscript.

For your convenience, we marked all changes in the manuscript in red.

We hope you find this revision has improved the manuscript and further consider it for publication. 

Kind regards, on behalf of the authors,

Reviewer: 3

28. The authors conduct a study in which they investigate possible ageism related to DT in physical therapy and find additional personal and social characteristics that possibly influence their attitudes toward wards using DT with older adults. From my point of view, I think it is an interesting and current issue due to the digital world in which we live and it is necessary to know if there are prejudices when using technology because it is important that the whole society can benefit from its use. Despite this, I believe that the work has many weaknesses and could be improved.

Thank you for acknowledging the relevance of the topic of our paper in a time of a more and more digitalized society. We have revised the paper according to your and the other 2 reviewers’ comments and hope that these changes address your points.

29. List of authors and affiliation: The e-mail address of each author must be provided

Thank you for noticing. We added the emails of all authors.

30. Abstract: The abstract should be a single paragraph and should follow the style of structured abstracts, but without headings: 1) Background, 2) Methods,3) Results, 4) Conclusion.

The abstract indeed follows these guidelines and structure of a single paragraph with no headings. We made sure that all sections 1) Background, 2) Methods 3) Results 4) Discussion and Conclusion are included.

31. Introduction: DT or DTs?

Thank you for noticing the unclarity. We have screened the entire article for inconsistencies and changed where needed. We use the abbreviation DT for digital technology as a general topic. When we mention DTs it means that we talk about digital technologies in terms of multiple devices that could be used.

32. Introduction: The study highlights that this issue may be further compounded by gender, as older women may be more likely to experience ageism in their interactions with DT. Authors should give the reference of the study. Is it Choi et al.

Thank you for pointing this out. Indeed, it is Choi et al. we refer to and we added this reference accordingly.

33. Introduction: Chang et al. found in their systematic review that 84.6% of the 56 studies they included showed that chronological age dictates which treatment or procedure will be chosen [16], meaning that a younger patient has a better outlook on receiving the gold standard treatment. Authors should place the reference after the author's name. Chang et al. [16]

Thank you for pointing this out, we have changed the place of the reference accordingly.

34. Introduction: For example, the systematic review looks at patients awaiting breast cancer surgeries and found in multiple studies that older patients were much more likely to be rejected for treatment than younger patients with identical histories. Rejected from what treatment?

Thank you for pointing out this unclarity. The paper we referenced to (Chang et al. [16]) indicates the following: ‘As an example, using vignettes of patients awaiting breast-cancer surgeries, a series of studies found that compared to younger patients with matched histories, older patients were significantly more likely to be denied treatment by new and advanced medical students as well as surgical and internal medicine residents’.
Based on this we refer to the surgical treatment that they were rejected from. To make this clearer to reader of our paper we specifically
added ‘surgical treatment’ in our sentence.

35. The authors relied on a previous study to calculate the sample size. Why these values? An a-priori statistical power analysis for logistic regression was done to determine the needed sample size using G*Power 3.1.9.4. (Heinrich Heine University Düsseldorf). A minimum sample size of 77 was calculated using the following assumptions: α of 0.05, assumed power of 90%, explained variance of our primary independent variable with other independent variables of 0.1 and the default odds ratio of 2.33.

Thank you for this comment, we realize this was not clear enough. α of 0.05 and assumed power of 90% are quite standard and common definitions for power analysis. We added an explanation that for the explained variance of our primary independent variable with other independent variables ‘we followed the G*Power recommended default settings’.

36. Methodology: A section on the type of study is missing

Thank you for pointing this out, we added a sub-section about study design in the method section, also in accordance with comment 3 of reviewer 1.

37. Methodology: In this section, I suggest to put also the information of the ethics committee.

Thank you for this suggestion. In the previous version of the paper, information about the ethics committee was included in the methods. However, following comment 13 from reviewer 1, we have moved all details about ethical considerations to the end of the manuscript, according to the journal’s format suggestions.

38. Instrument: A Cronbach's α coefficient of .819 was found for the scale. Is there any reference data for this instrument? Personally, in this methodology section I would only put the reference values of Cronbach's. The values obtained in this study I would put in the reference values of Cronbach's. I would put those obtained in this study in the results section.

Thank you for this suggestion. The scale you are referring to is a scale that we modified from the TAM2. For clarity, we added a reference for the Cronbach’s alpha originally found for the TAM2 in the revised paper. The second part of your remark addressed the location in the paper. Cronbach’s alpha’s are sometimes described in the Tools section of the method or in the results section. As reliability of the scales was not the main goal of this paper, we prefer to leave it in the methods section, in order to keep the results more focused on the use of DT in treatment in older persons, which is the main focus of our study.

39. Produce: How did you get in touch with the sample? They were sent an invitation by mail but how did you get the mails? Through professional associations...?

Thank you for pointing this out. This was also suggested by reviewer 1 in comment 4. We modified the following in the manuscript: ‘Certified physiotherapists working in Luxembourg were recruited via an email invitation which was send out by the Luxembourgish Physiotherapy association (ALK), and by a convenience sample through the authors’ professional networks and social media.

40. Analysis: Was the assumption of normality tested? Results are expressed as means and SD. These statistics are used for normal samples.

Indeed, the assumption of normality was tested, however was not reported as it is not a required assumption for logistic regression.

41. Analysis: SPSS Statistics 28 (IBM US). Use the full reference. Suggestion: IBM SPSS Statistics for Windows, Version 28.0. Armonk, NY: IBM Corp.

Thank you. We have added more detail in the text.

42. Analysis: A correlation matrix was created for insight into the relation between all variables. by which statistical test? Pearson, Sperman?

We used Pearson correlations. We added this in the title.

43. Analysis: Were the applicability assumptions of logistic regression tested?

Thank you for pointing this out. Indeed, the assumptions of independence of errors, linearity in the logit for continuous variables, absence of multicollinearity, and lack of strongly influential outliers, were checked and tested, and were not found to be violated. We have added this explanation.

44. Results: The wording of the results should include the main results in terms of numbers (more next findings).

If we understand the reviewer correctly, the reviewer’s concern is that the results should also include  figures (apart from the tables). We reviewed the results section carefully and made sure that we reported all the main results as figures in the results section (i.e., percentages, means, statistical tests, p-values, etc).

45. Results: No table of descriptive statistics?

Thank you for noticing this. As we already have two additional tables in the results section, we chose to provide all descriptives of the characteristics in the text of the first paragraph of the results section. Additionally, descriptives of the scales used (means and SDs) are provided in table 1.

46. Results: Line 237: not is write in italics

Thank you for pointing this out, we changed it to non-italics.

47. Results: In the line 244: Standard Deviations (SD). This should appear the first time the abbreviation appears.

Thank you for this suggestion. We made sure that this is the case.

48. Results: Line 247: compared to previous results with physiotherapists [8]. In the results section, results cannot be discussed with other studies.

Thank you for pointing this out, we rephrased the sentence and removed the comparison with any previous study.

49. Results: Line 246: ATOAUT-11 scale was 44.29 (of max 66)? because no SD is given here?

We have added the SD in the text as well as in Table 1.

50. Results: Correlations with qualitative data?

We did not report any correlations with qualitative data. All variables in the correlation matrix are continuous or dummy coded (0,1).

If we understand correctly, the reviewer might be referring to a similar issue raised by reviewer 1 in comment 15 (which is not related to the correlation matrix), to which we replied: We used a yes or no question to ask if or if not the participant did not offer DT-based treatment to a patient because of their age. Following, we asked in a second question – when they answered yes - about the reasons why. This question was a multiple response question with a closed list of possible responses (and participants could select more than one category). This is clarified now in the ‘Tools’ section.

51. Results: + Correlation is significant at the 0.1 level (2-tailed)?

Thank you for this question. We indicated 0.1 level significance in the table to allow the reader to see also weak and close to 0.05 correlations. Nevertheless, in the text we only referred to 0.05 as a significant correlation. We therefore removed the 0.1 level from the table.

52. Results: However, the model was only marginally significant at a p=0.058 level. This is not significant. It should be < 0.05. The authors also put it in the analysis section:P-values <0.05 were considered statistically significant for all analyses

Thank you for pointing this out. “marginally” is used in this case to indicate that it is close to significant. In our experience, this term is often used. However, following your suggestion, we adapted this and instead of “marginally significant” we now say “did not reach statistical significance (p = 0.058)”.

53. Results: Table 1. Should be reorganized in the tables, all abbreviations should be described in the table footnote.

Thank you for pointing this out, we reorganized the table and made sure that all the abbreviations are in the footnotes of the table.

54. Discussion: Could be improved

We have made several changes in the discussion. Please see our answer to the comments 17-19 from reviewer 1.

55. Conclusion: It should be specified that the findings are in terms of the sample studied.

Thank you for this comment. We now more clearly specified the limitations of the sample in the limitations section of the discussion. As per comment 20 also from reviewer 1, we also have revised the whole conclusion section. We believe it now reflects your concern as well as other comments raised about the content of the conclusion.

56. Author Contributions: I think this section is not filled in.

Thank you for noticing, we filled this online, but this was indeed not added to the version you received. We have added this section.

57. Funding: I think this section is not filled in.

Thank you for your suggestion. We have moved this from the ‘Acknowledgements’ to the ‘Funding’ section.

58. Informed Consent Statement: I think this section is not filled in.

Thank you for your suggestion. We have added content to the appropriate section according to the journal format.

Round 2

Reviewer 1 Report

Dear authors, 

I would like to congratulate you on the changes made and the improvement of the article. 

Best regards, 

Author Response

We would like to thank the reviewers for their time to check the previous revision.

We are happy that we were able to address most comments by the reviewers and thank them for their additional feedback on the current manuscript.

We thank reviewer 1 for approving the revisions.

We provided below our response to the remining comments by reviewer 3. We used the original numbering of the comments for convenience. All changes in the manuscript are marked in red.

Kind regards on behalf of the authors.

Reviewer: 3

 I would like to congratulate the authors since, in my opinion; they have clarified doubts that may have arisen for the readers. I consider that the article has improved its quality. However, there are some minor issues that I would like to comment on and that you may consider if you think it appropriate.

Thank you for this positive feedback.

4) Thanks to the authors for considering my comment. I knew what each acronym meant; my comment was because you define digital technology (DT) and yet indistinctly use either DT or DTS.

We have changed all mentionings to DT in the current revision.

8) Thank you for your answer even though I think I did not explain myself correctly. My comment was not directed exactly to the explanation of how a sample calculation is made. The comment was directed precisely to the fact that if the authors relied on a previous study to take the reference data, in my opinion this should be referenced.

Thank you for clarifying. We have added the following reference by Faul et al. for using G power for logistic regression, and made the explanation more consise.

Faul, F.; Erdfelder, E.; Buchner, A.; Lang, A. G. Statistical power analyses using G* Power 3.1: Tests for correlation and regression analyses. Behavior research methods 2009, 41, 1149-1160.

10) Thank you for taking my comment into account. The authors in the previous version mentioned the ethical considerations at the end of the methodology section. I personally think it would be more correct at the beginning of this section, after the study design. However, and following the recommendations of R1, I think it is correct.

Thank you. We indeed decided to follow the journal format as suggested by reviewer 1.

11) Thank you for your response: My comment is because in my opinion it is not correct to mention an own result in the methods section. It talks before the result and however if the authors describe in the analysis section (later) that they are going to perform this type of analysis. In my opinion, the results section should show the results of all the tests performed.

As per your request, we have moved the report of Cronbach’s alpha to the results in line 278: “Cronbach’s alpha for all scales were respectively high and comparable with pre-vious literature, suggesting good reliability of the scales used (.819 for the Attitudes Towards Technology Use in the Work Environment; .845 for the ATOAUT scale; and .791 for the ERA-12 scale).”

13) Thank you for your response. Obviously, normality is not necessary for logistic regression but the assumption of normality is necessary to use parametric tests like Pearson and to use descriptive means and standard deviations.

We have added the following in line 263, when discussing the variables in table 1: “Examining skewness and kurtosis values revealed no extreme violations of normality.”

Reviewer 3 Report

Reviewer: 3

I would like to congratulate the authors since, in my opinion; they have clarified doubts that may have arisen for the readers. I consider that the article has improved its quality. However, there are some minor issues that I would like to comment on and that you may consider if you think it appropriate.

1.     The authors conduct a study in which they investigate possible ageism related to DT in physical therapy and find additional personal and social characteristics that possibly influence their attitudes toward wards using DT with older adults. From my point of view, I think it is an interesting and current issue due to the digital world in which we live and it is necessary to know if there are prejudices when using technology because it is important that the whole society can benefit from its use. Despite this, I believe that the work has many weaknesses and could be improved.

Thank you for acknowledging the relevance of the topic of our paper in a time of a more and more digitalized society. We have revised the paper according to your and the other 2 reviewers’ comments and hope that these changes address your points.

2.     List of authors and affiliation: The e-mail address of each author must be provided

Response review:

Thanks

Thank you for noticing. We added the emails of all authors.

3.     Abstract: The abstract should be a single paragraph and should follow the style of structured abstracts, but without headings: 1) Background, 2) Methods,3) Results, 4) Conclusion.

Response review:

Thanks

The abstract indeed follows these guidelines and structure of a single paragraph with no headings. We made sure that all sections 1) Background, 2) Methods 3) Results 4) Discussion and Conclusion are included.

4.     Introduction: DT or DTs?

Thank you for noticing the unclarity. We have screened the entire article for inconsistencies and changed where needed. We use the abbreviation DT for digital technology as a general topic. When we mention DTs it means that we talk about digital technologies in terms of multiple devices that could be used.

Response review:

Thanks to the authors for considering my comment. I knew what each acronym meant; my comment was because you define digital technology (DT) and yet indistinctly use either DT or DTS.

5.     Introduction: The study highlights that this issue may be further compounded by gender, as older women may be more likely to experience ageism in their interactions with DT. Authors should give the reference of the study. Is it Choi et al.

Thank you for pointing this out. Indeed, it is Choi et al. we refer to and we added this reference accordingly.

Response review:

Thanks

6.     Introduction: Chang et al. found in their systematic review that 84.6% of the 56 studies they included showed that chronological age dictates which treatment or procedure will be chosen [16], meaning that a younger patient has a better outlook on receiving the gold standard treatment. Authors should place the reference after the author's name. Chang et al. [16]

Thank you for pointing this out, we have changed the place of the reference accordingly.

Response review:

Thanks

7.     Introduction: For example, the systematic review looks at patients awaiting breast cancer surgeries and found in multiple studies that older patients were much more likely to be rejected for treatment than younger patients with identical histories. Rejected from what treatment?

Thank you for pointing out this unclarity. The paper we referenced to (Chang et al. [16]) indicates the following: ‘As an example, using vignettes of patients awaiting breast-cancer surgeries, a series of studies found that compared to younger patients with matched histories, older patients were significantly more likely to be denied treatment by new and advanced medical students as well as surgical and internal medicine residents’.

Based on this we refer to the surgical treatment that they were rejected from. To make this clearer to reader of our paper we specifically
added ‘surgical treatment’ in our sentence.

Response review:

Thanks

8.      The authors relied on a previous study to calculate the sample size. Why these values? An a-priori statistical power analysis for logistic regression was done to determine the needed sample size using G*Power 3.1.9.4. (Heinrich Heine University Düsseldorf). A minimum sample size of 77 was calculated using the following assumptions: α of 0.05, assumed power of 90%, explained variance of our primary independent variable with other independent variables of 0.1 and the default odds ratio of 2.33.

Thank you for this comment, we realize this was not clear enough. α of 0.05 and assumed power of 90% are quite standard and common definitions for power analysis. We added an explanation that for the explained variance of our primary independent variable with other independent variables ‘we followed the G*Power recommended default settings’.

Response review:

Thank you for your answer even though I think I did not explain myself correctly. My comment was not directed exactly to the explanation of how a sample calculation is made. The comment was directed precisely to the fact that if the authors relied on a previous study to take the reference data, in my opinion this should be referenced.

9.     Methodology: A section on the type of study is missing

Response review:

Thanks

Thank you for pointing this out, we added a sub-section about study design in the method section, also in accordance with comment 3 of reviewer 1.

10.  Methodology: In this section, I suggest to put also the information of the ethics committee.

Thank you for this suggestion. In the previous version of the paper, information about the ethics committee was included in the methods. However, following comment 13 from reviewer 1, we have moved all details about ethical considerations to the end of the manuscript, according to the journal’s format suggestions.

Response review:

Thank you for taking my comment into account. The authors in the previous version mentioned the ethical considerations at the end of the methodology section. I personally think it would be more correct at the beginning of this section, after the study design. However, and following the recommendations of R1, I think it is correct.

11.  Instrument: A Cronbach's α coefficient of .819 was found for the scale. Is there any reference data for this instrument? Personally, in this methodology section I would only put the reference values of Cronbach's. The values obtained in this study I would put in the reference values of Cronbach's. I would put those obtained in this study in the results section.

Thank you for this suggestion. The scale you are referring to is a scale that we modified from the TAM2. For clarity, we added a reference for the Cronbach’s alpha originally found for the TAM2 in the revised paper. The second part of your remark addressed the location in the paper. Cronbach’s alpha’s are sometimes described in the Tools section of the method or in the results section. As reliability of the scales was not the main goal of this paper, we prefer to leave it in the methods section, in order to keep the results more focused on the use of DT in treatment in older persons, which is the main focus of our study.

Response review:

Thank you for your response: My comment is because in my opinion it is not correct to mention an own result in the methods section. It talks before the result and however if the authors describe in the analysis section (later) that they are going to perform this type of analysis. In my opinion, the results section should show the results of all the tests performed.

12.  Produce: How did you get in touch with the sample? They were sent an invitation by mail but how did you get the mails? Through professional associations...?

Thank you for pointing this out. This was also suggested by reviewer 1 in comment 4. We modified the following in the manuscript: ‘Certified physiotherapists working in Luxembourg were recruited via an email invitation which was send out by the Luxembourgish Physiotherapy association (ALK), and by a convenience sample through the authors’ professional networks and social media.

Response review:

Thanks

13.  Analysis: Was the assumption of normality tested? Results are expressed as means and SD. These statistics are used for normal samples.

Indeed, the assumption of normality was tested, however was not reported as it is not a required assumption for logistic regression.

Response review:

Thank you for your response. Obviously, normality is not necessary for logistic regression but the assumption of normality is necessary to use parametric tests like Pearson and to use descriptive means and standard deviations.

14.  Analysis: SPSS Statistics 28 (IBM US). Use the full reference. Suggestion: IBM SPSS Statistics for Windows, Version 28.0. Armonk, NY: IBM Corp.

Thank you. We have added more detail in the text.

Response review:

Thanks

15.  Analysis: A correlation matrix was created for insight into the relation between all variables. by which statistical test? Pearson, Sperman?

We used Pearson correlations. We added this in the title.

Response review:

Thanks

16.  Analysis: Were the applicability assumptions of logistic regression tested?

Thank you for pointing this out. Indeed, the assumptions of independence of errors, linearity in the logit for continuous variables, absence of multicollinearity, and lack of strongly influential outliers, were checked and tested, and were not found to be violated. We have added this explanation.

Response review:

Thanks

17.  Results: The wording of the results should include the main results in terms of numbers (more next findings).

If we understand the reviewer correctly, the reviewer’s concern is that the results should also include  figures (apart from the tables). We reviewed the results section carefully and made sure that we reported all the main results as figures in the results section (i.e., percentages, means, statistical tests, p-values, etc).

Response review:

Thanks

18.  Results: No table of descriptive statistics?

Thank you for noticing this. As we already have two additional tables in the results section, we chose to provide all descriptives of the characteristics in the text of the first paragraph of the results section. Additionally, descriptives of the scales used (means and SDs) are provided in table 1.

Response review:

Thanks

19.  Results: Line 237: not is write in italics

Thank you for pointing this out, we changed it to non-italics.

Response review:

Thanks

20.  Results: In the line 244: Standard Deviations (SD). This should appear the first time the abbreviation appears.

Thank you for this suggestion. We made sure that this is the case.

Response review:

Thanks

21.  Results: Line 247: compared to previous results with physiotherapists [8]. In the results section, results cannot be discussed with other studies.

Thank you for pointing this out, we rephrased the sentence and removed the comparison with any previous study.

Response review:

Thanks

22.  Results: Line 246: ATOAUT-11 scale was 44.29 (of max 66)? because no SD is given here?

We have added the SD in the text as well as in Table 1.

Response review:

Thanks

23.  Results: Correlations with qualitative data?

We did not report any correlations with qualitative data. All variables in the correlation matrix are continuous or dummy coded (0,1).

If we understand correctly, the reviewer might be referring to a similar issue raised by reviewer 1 in comment 15 (which is not related to the correlation matrix), to which we replied: We used a yes or no question to ask if or if not the participant did not offer DT-based treatment to a patient because of their age. Following, we asked in a second question – when they answered yes - about the reasons why. This question was a multiple response question with a closed list of possible responses (and participants could select more than one category). This is clarified now in the ‘Tools’ section.

Response review:

Thanks

24.  Results: + Correlation is significant at the 0.1 level (2-tailed)?

Thank you for this question. We indicated 0.1 level significance in the table to allow the reader to see also weak and close to 0.05 correlations. Nevertheless, in the text we only referred to 0.05 as a significant correlation. We therefore removed the 0.1 level from the table.

Response review:

Thanks

25.  Results: However, the model was only marginally significant at a p=0.058 level. This is not significant. It should be < 0.05. The authors also put it in the analysis section: P-values <0.05 were considered statistically significant for all analyses

Thank you for pointing this out. “marginally” is used in this case to indicate that it is close to significant. In our experience, this term is often used. However, following your suggestion, we adapted this and instead of “marginally significant” we now say “did not reach statistical significance (p = 0.058)”.

Response review:

Thanks

26.  Results: Table 1. Should be reorganized in the tables, all abbreviations should be described in the table footnote.

Thank you for pointing this out, we reorganized the table and made sure that all the abbreviations are in the footnotes of the table.

Response review:

Thanks

27.  Discussion: Could be improved 

We have made several changes in the discussion. Please see our answer to the comments 17-19 from reviewer 1.

Response review:

Thanks

28.  Conclusion: It should be specified that the findings are in terms of the sample studied. 

Thank you for this comment. We now more clearly specified the limitations of the sample in the limitations section of the discussion. As per comment 20 also from reviewer 1, we also have revised the whole conclusion section. We believe it now reflects your concern as well as other comments raised about the content of the conclusion.

Response review:

Thanks

29.  Author Contributions: I think this section is not filled in.

Thank you for noticing, we filled this online, but this was indeed not added to the version you received. We have added this section.

Response review:

Thanks

30.  Funding: I think this section is not filled in.

Thank you for your suggestion. We have moved this from the ‘Acknowledgements’ to the ‘Funding’ section.

Response review:

Thanks

31.  Informed Consent Statement: I think this section is not filled in.

Thank you for your suggestion. We have added content to the appropriate section according to the journal format.

Response review:

Thanks

Author Response

We would like to thank the reviewers for their time to check the previous revision.

We are happy that we were able to address most comments by the reviewers and thank them for their additional feedback on the current manuscript.

We provided below our response to the remining comments by reviewer 3. We used the original numbering of the comments for convenience. All changes in the manuscript are marked in red.

Kind regards on behalf of the authors.

Reviewer: 3

 I would like to congratulate the authors since, in my opinion; they have clarified doubts that may have arisen for the readers. I consider that the article has improved its quality. However, there are some minor issues that I would like to comment on and that you may consider if you think it appropriate.

Thank you for this positive feedback.

4) Thanks to the authors for considering my comment. I knew what each acronym meant; my comment was because you define digital technology (DT) and yet indistinctly use either DT or DTS.

We have changed all mentionings to DT in the current revision.

8) Thank you for your answer even though I think I did not explain myself correctly. My comment was not directed exactly to the explanation of how a sample calculation is made. The comment was directed precisely to the fact that if the authors relied on a previous study to take the reference data, in my opinion this should be referenced.

Thank you for clarifying. We have added the following reference by Faul et al. for using G power for logistic regression, and made the explanation more consise.

Faul, F.; Erdfelder, E.; Buchner, A.; Lang, A. G. Statistical power analyses using G* Power 3.1: Tests for correlation and regression analyses. Behavior research methods 2009, 41, 1149-1160.

10) Thank you for taking my comment into account. The authors in the previous version mentioned the ethical considerations at the end of the methodology section. I personally think it would be more correct at the beginning of this section, after the study design. However, and following the recommendations of R1, I think it is correct.

Thank you. We indeed decided to follow the journal format as suggested by reviewer 1.

11) Thank you for your response: My comment is because in my opinion it is not correct to mention an own result in the methods section. It talks before the result and however if the authors describe in the analysis section (later) that they are going to perform this type of analysis. In my opinion, the results section should show the results of all the tests performed.

As per your request, we have moved the report of Cronbach’s alpha to the results in line 278: “Cronbach’s alpha for all scales were respectively high and comparable with pre-vious literature, suggesting good reliability of the scales used (.819 for the Attitudes Towards Technology Use in the Work Environment; .845 for the ATOAUT scale; and .791 for the ERA-12 scale).”

13) Thank you for your response. Obviously, normality is not necessary for logistic regression but the assumption of normality is necessary to use parametric tests like Pearson and to use descriptive means and standard deviations.

We have added the following in line 263, when discussing the variables in table 1: “Examining skewness and kurtosis values revealed no extreme violations of normality.”